# Electrochemical Sensor for Bilirubin Detection Using Paper-Based Screen-Printed Electrodes Functionalized with Silver Nanoparticles

**DOI:** 10.3390/mi13111845

**Published:** 2022-10-28

**Authors:** Nigar Anzar, Shariq Suleman, Rocky Kumar, Rachna Rawal, Chandra Shekhar Pundir, Roberto Pilloton, Jagriti Narang

**Affiliations:** 1Department of Biotechnology, Jamia Hamdard University, New Delhi 110062, India; 2Department of Physics and Astrophysics, University of Delhi, Delhi 110007, India; 3Department of Biochemistry, Maharshi Dayanand University, Rohtak 124001, India; 4Institute of Crystallography of National Research council (IC-CNR), Monterotondo, I-00015 Rome, Italy

**Keywords:** bilirubin, electrochemical, serum, nanoparticles, jaundice

## Abstract

A notable diagnostic for the detection of hemolytic diseases is bilirubin, a by-product of haemoglobin breakdown. The concentration of bilirubin ranges from 0.3 to 1.9 mg in 100 mL of blood. Low blood bilirubin levels are associated with a greater risk of coronary heart disease and anaemia. Hyperbilirubinemia results from a serum bilirubin level of more than 2.5 mg/100 mL. Therefore, it is very crucial to check the serum bilirubin level. Analytical equipment for point-of-care testing must be portable, small, and affordable. A unique method is used to detect bilirubin selectively using paper-based screen-printed carbon electrodes that were covalently linked with nanoparticles, that serves as a key biomarker for jaundice. In order to create an electrochemical biosensor, bilirubin oxidase was immobilised on electrodes modified with AgNPs. The morphology of Ag nanoparticles in terms of size and shape was determined using both UV- Vis Spectroscopy and transmission electron microscopy (TEM). The biosensor’s analytical response was assessed using potentiostat (Cyclic voltammetry (CV) and linear sweep voltammetry (LSV)). The developed paper-based sensor provided optimum feedback and a broad linear range of 1 to 9 µg/mL for bilirubin, with a lower LOD of 1 µg/mL. Through tests of bilirubin in artificial blood serum, the viability is confirmed. The method that is being used makes it possible to create and use an inexpensive, miniature electrochemical sensor.

## 1. Introduction

A residue of the natural breakdown of aged red blood cells is called bilirubin (BR), a pigment yellow in colour, which, being a catabolic process, is required to eliminate waste. Unconjugated (Bu) and conjugated (Bc) bilirubin, are the two types of bilirubin found in human serum [1]. Conjugated BR and glucuronic acid combine to generate a water-soluble compound while the unconjugated free BR bonds with albumin and becomes water soluble [2]. A little amount of free BR that isn’t bound to albumin is a crucial marker of bilirubin poisoning. Direct bilirubin levels in human blood are typically between 1–5 μM (0.06–0.3 mg/dL) and Around 25 μM (1.23 mg/dL) of BR is present totally. High bilirubin levels can indicate a number of illnesses, including gallstones, jaundice, diarrhoea, liver dysfunction, hemolytic anaemia, etc. [3]. Hyperbilirubinemia is the outcome of an excessive bilirubin concentration. Bilirubin deposits on numerous tissues as a result of hyperbilirubinemia, producing metabolic problems that can result in jaundice or icterus, hepatitis, mental health concerns, cerebral palsy, brain damage, and even death (exclusively in the case of neonates) [4]. Jaundice’s primary signs and symptoms include yellowing of the skin and eyes, pale urine, itchy skin, lethargy, and poor appetite [5]. Many newborns experience newborn jaundice, which is caused by elevated bilirubin levels within a few days of delivery. Newborns have more red blood cells and breakdown them more quickly, which is why it occurs; however, their liver are still not developed enough to manage this. In order to avoid newborn jaundice, serum bilirubin content is therefore evaluated in neonates 24 h old in the majority of hospitals. Although coronary heart disease and iron insufficiency are linked to decreased bilirubin concentration [6]. The clinical importance of bilirubin measurement is crucial for the identification, prognosis, and management of hemolytic diseases in both adults and children. Therefore, it is crucial to create new techniques that can precisely measure the bilirubin content in biological fluid [7].

Direct spectroscopic measurements and the diazo reaction are the most popular ways to detect BR among the various techniques that have been developed. Other proteins can interfere with spectroscopic methods, and the accuracy of the diazo method is uncertain because of the pH dependence of the reaction rate [8]. Method of chemiluminescence, in which bilirubin is subjected to redox reactions in aqueous solutions to produce chemiluminescence [9]. The disadvantages of this approach are its lack of adequate selectivity and sensitivity to a of physiochemical variables [10]. Polarography, fluorometry, and colorimetric assay kits are a few more well-known and commercially accessible techniques, although they are more costly, require more skill, and are regarded as less sensitive [11]. These techniques are not effective for routine testing and detection due to the above-mentioned drawbacks. However, bilirubin oxidase (BOx) based biosensors are easy to use since no sample pre-treatment is necessary. They are also quick and highly sensitive. These biosensors work by either oxidizing hydrogen peroxide or consuming oxygen. When there is molecular oxygen present, BOx converts the BR into biliverdin, and by monitoring the loss of molecular oxygen or the production of hydrogen peroxide, one might estimate the concentration of BR [12,13]. H_2_O_2_ generated from the Box, splits up into protons, oxygen and electrons under the presence of applied voltage. Thus, producing electrochemical energy from chemical energy [14]. Reaction involved are as follows:
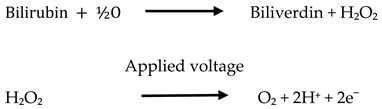


Since the last decade, the use of nanomaterials to enhance biosensor performance has become incredibly popular. The use of nanoparticles in biosensing is believed to be ground-breaking and powerful, because of the distinctive size-dependent properties of nanoscale materials, which have had a significant impact on many facets of human existence, nanotechnology is a promising issue for biomedical applications. Metal nanoparticles, like silver, which have exceptional optical, electrical, and antibacterial properties, have garnered a lot of interest over the years [15,16]. Screen printed carbon electrodes (SPEs) have recently been shown to be an effective substitute for pricey traditional electrodes [17,18]. They not only meet the requirements for cheap cost and compact size, but they also offer practical benefits such low detection limit, high adaptability, quick reaction time, high repeatability, and have tremendous potential for mass manufacturing and downsizing [19].

Hence, in the present study, we have chosen Screen printed paper electrodes to develop a point-of-care assay for bilirubin detection. Silver nanoparticles are especially attractive for sensor fabrication as they have fascinating properties, such as large surface area, good mechanical and electrical properties, and excellent chemical/thermal stability [20]. Thus, we have deposited silver nanoparticles on developed paper-based electrodes and studied the resulting enhancement of the electroanalytical parameters, such as lower detection limit, wider linear range, and higher sensitivity. The viability of the sensor was validated by measuring the BR concentration in the artificial human serum sample as well.

## 2. Materials and Methods

### 2.1. Chemical, Reagents and Apparatus

All the chemicals were of AR grade. For the fabrication of electrodes: black carbon conductive ink and silver conductive ink paste was purchased from SNAB GRAPHIX (Bengaluru, INDIA) PVT Ltd. For the preparation of PBS: NaCl, KCl, Na_2_HPO_4_, and KH_2_PO_4_ were purchased from LOBA. Bilirubin, Bilirubin Oxidase, Potassium ferricyanide, potassium ferrocyanide were purchased from MTOR Lifescience pvt ltd. For Artificial serum: NaCl, KCL, NaHCO_3_, NaH_2_PO_4_, CaCl_2_, MgSO_4_, Na_2_HPO_4_ were used. To synthesize Nanoparticles (silver): Silver nitrate (Qualigens, Mumbai, India), ethanol, sodium Hydroxide, Sodium Borohydride were used of AR grade.

Metrohm Dropsens (Stat-I 400s, Metrohm AG, Herisau, Switzerland) was utilized for electrochemical measurements such as cyclic voltammetry (CV) and linear sweep voltammetry (LSV). Silver nanoparticle morphology was inspected via Transmission electron microscopy (TEM) on Talos L120C (Thermo Fisher Scientific, Waltham, MA, USA). UV-Vis absorbance was measured using Agilent technologies, Cary100 series, and UV-Vis spectrometer (Agilent Technologies, Santa Clara, CA, USA).

### 2.2. Synthesis of Silver Nanoparticles

For silver nanoparticle synthesis, chemical method was employed [21]. In this, two chemicals were used, i.e., silver nitrate (10 mL of 1.0 mM) & sodium borohydride (30 mL of 2.0 mM). Then, the solution of both chemicals was prepared, and after that, silver nitrate was added dropwise to the ice-cold sodium borohydride solution while stirring. The solution’s color turns bright yellow, indicating the successful synthesis of silver nanoparticles.

### 2.3. Development of Three Electrodes System-Based Biosensor

A silk screen with a laser-cut patterned solid skin attached to it and predetermined dimensions for a three-electrode arrangement was employed for hand printing of electrodes on white paper sheets. Carbon conductive ink was squeezed onto the sheets using a squeezer through the defined overhead screen’s open areas. The dimensions of the electrode were pre-established and framed on the silk screen, which was then employed as a stencil for the electrode preparation. The printed electrodes consisted of three electrodes: a counter electrode (CE), a working electrode (WE), and a reference electrode (RE) drop casted with Ag/AgCl. This resulted in the manufacture of three-electrode system-based biosensors. (As shown in Figure 1).

### 2.4. Deposition of the Silver Nanoparticle and Immobilization on the Paper-Based Sensor

The synthesized silver nanoparticle was drop-deposited (30 µL) on the circular working area of the paper electrode. The paper electrode was then dried for overnight. After modification with silver nanoparticle, the working area was immobilized with the enzyme (Bilirubin Oxidase). For this, 20 µL of enzyme was dropped over the working area. These nanoparticle/enzyme-modified electrodes were further used for the detection of bilirubin.

### 2.5. Stages for Electrochemical Detection

To make a functioning and selective biosensor for detecting bilirubin, the enzyme (Bilirubin Oxidase) needs to be deposited onto the working electrode with the help of nanoparticle. For this, CV and LSV values of bare electrodes, with no deposition, were analyzed. Next, the Ag NPs were deposited onto the paper-based sensor and dried overnight, after which both voltammetry’s were repeated as before. For the next step, the enzyme was deposited on the paper-based sensor containing dried Ag NPs and CV/LSV values were recorded. For the last phase of paper-based sensor design, the analyte molecule bilirubin was added onto electrodes containing both Ag NPs and enzyme, and CV/LSV were performed and recorded with the help of Ferro/Ferri in PBS (Figure 2).

### 2.6. Binding of the Analyte on Enzyme/Ag NPs/PBs and Optimisation of Physio-Chemical Parameters

The paper-based sensors detection capability was adjusted by examining the variations in the voltammograms produced by changes in the different detection settings. This was accomplished by adjusting the parameters of the bilirubin sample. Concentrations of 1, 3, 5, 7, 9 µg/mL, were prepared, bilirubin was incubated at diverse temperatures (15, 25, 35, and 45 °C), and the time given for interaction between enzyme and substrate by the potentiostat (tcond) was changed. The bilirubin concentrations (1, 3, 5, 7 and 9 µg/mL) were determined by dropping a combination of antigen and ferro/Ferri in PBS. The CV/LSV measurements were performed to confirm the interaction of the paper-based sensor and bilirubin. Bilirubin was added onto the working surface of enzyme/AgNPs Paper based sensor at various concentrations.

### 2.7. Procedure for Real Sample Analysis, Repeatability, and Stability Analysis

The sensor’s ability to operate in actual samples was tested by adding a fixed concentration of bilirubin to artificial human serum. The electrochemical evaluations were performed to validate the results. The concentration of bilirubin was repeated numerous times, demonstrating the repeatability of the proposed biosensor, and its stability was verified for at least a month.

## 3. Results and Discussion

The electrode fabrication involves the deposition of a silver nanoparticle on the surface of a bare working electrode. The silver nanoparticle is drop-deposited onto the electrode’s surface. Enzyme was also applied on working electrodes after drop deposition of silver nanoparticles. The silver nanoparticle offers a biocompatible environment for the biological identifying component while also accelerating electron transmission. This biosensor is mainly dependent on the nanomaterial. Therefore, the synthesis and characterization of these nanomaterials are one of the crucial steps of the study.

### 3.1. Characterization of Silver Nanoparticles

The successful synthesis of silver nanoparticle was confirmed by using the following characterization techniques—TEM and UV-vis spectroscopy, the most significant technique to confirm the formation of nanomaterials. Figure 3a shows UV–visible spectrum of the aqueous medium containing silver nanoparticles absorption peak at around 400 nm. Figure 3b depicts the TEM image that shows clear spherical nanoparticles that are well-dispersed with an average size of about 40 nm.

### 3.2. Electro-Chemical Properties of Bilirubin/Enzyme/AgNPs/PBs

The electrochemical characterization of Bilirubin/enzyme/Ag NPs/PBs modified electrodes was performed by employing the electrochemical techniques like cyclic voltammetry (CV) and linear sweep voltammetry (LSV) using Potentiostat. Figure 4 shows the differential current response at different stages of the electrode validated by both CV and LSV. In CV- bare electrode (ePAD) showed a smaller peak current response which is attributed to the less electron transfer kinetics. Upon deposition of silver nanoparticles onto the working surface, there was a significant increase in current response, by two-fold, which is due to the fast electron transfer kinetics provided by silver nanoparticles. After immobilization of the enzyme (Box) onto the working surface, the current gets drastically decreased due to the non-conducive nature of the enzyme. After the introduction of Bilirubin, the current response was further increased as compared to enzyme (Box).

In LSV-the bare electrode (ePAD) showed a smaller peak current response, which is attributed to the less electron transfer kinetics. Upon deposition of silver nanoparticles onto the working surface, there was a significant increase in current response by two-fold, which is due to the fast electron transfer kinetics provided by silver nanoparticles. After immobilization of the enzyme (BOx) onto the working surface, the current gets drastically decreased due to the non-conducive nature of enzyme. After the introduction of target molecule (Bilirubin), the current response increases as compared to enzyme (BOx).

### 3.3. Effect of the Different Bilirubin Concentrations on the Enzyme/Ag-NPs/PBs and Optimization of Bilirubin/Enzyme/Ag-NPs/PBs in Terms of Temperature and Time

Different concentrations of bilirubin were analysed to depict the quantitative performance of the developed sensor. Different concentrations varying from 1, 3, 5, 7, 9 µg/mL were employed. The results concluded that bilirubin is showing interaction with enzyme and at different concentrations varying current response was observed which confirmed the quantitative performance of the developed sensor. The results obtained were in line with the earlier reported sensors. Upon increasing the concentration of bilirubin there was an increase in current response due to the enhancement of oxidation current achieved by increased concentration of H_2_O_2_ during the enzymatic reaction. The produced H_2_O_2_ under the applied voltage splits into oxygen, protons and electrons. Therefore, the overall chemical energy gets converted into electrical energy. The detection limit was found to be 1 µg/mL. The concentration results were validated via two electrochemical measurements, i.e., CV and LSV. (Figure 5).

The optimization of the biosensor is a critical step for the proper functioning of the designed sensor. The sensor’s performance is affected by temperature and time. As a result, the sensor was adjusted in terms of these experimental factors to obtain maximum responsiveness. The performance of the developed sensor bilirubin/enzyme/Ag-NPs/ePAD/PBs was extensively studied in varying ranges of temperature and time. Cyclic voltammogram was observed at different temperatures ranging from 15 °C to 55 °C at a scan rate of 50 mV/s. The highest response was observed at 45 °C, but it was near to the enzyme. Therefore, the sensor was optimized at 35 °C. The developed sensor was optimized at different times (seconds) so that a maximum response can be observed at a time. (Figure 6).

### 3.4. Detection Limit and Precision/Accuracy (Recovery) Test of the Paper-Based Sensor

The limit of detection (LOD) is defined as the least concentration of the target that could be detected with the electrode, resulting in a LOD of 1 µg/mL. To assess precision and accuracy, a recovery test was done. Consequently, different bilirubin concentrations have been spiked into the other concentration. Therefore, 1 µg/mL bilirubin concentration was added to the other concentration, which shows the current almost equal to the concentration of 3 µg/mL demonstrating the great recovery of the proposed biosensor. The CV graph of recovery is depicted in Figure 7.

### 3.5. Analysis of Specificity/Reliability (Cross-Reactivity) and Stability

Urea and Albumin (1µg/mL) were used to test the cross-reactivity performance of the paper-based sensor. Figure depicts the current responses of the detection samples recorded with CV. Figure 6 shows that the peak flow of Urea and Albumin is nearly identical to that of the Enzyme/Ag-NPs/PBs. Furthermore, the paper-based sensor was maintained at 4 °C for various days (1 day, 7 days, 15 days, and 30 days) to identify bilirubin (1µg/mL) and examine the stability of the paper-based sensor using the cyclic voltammetry technique. The findings clearly demonstrated that the built sensor was stable until the 15th day and produced almost identical results to enzyme/Ag NPs/PBs after the 15th day, as shown in (Figure 8). The stability graph also demonstrated the error bar, which shows enough reproducibility and repetition for bilirubin detection.

### 3.6. Analysis of Bilirubin in Human Artificial Serum

Bilirubin (1 μg/mL) was also tested on artificial human serum. The results were as similar to the previous ones. The developed sensor was capable of detecting the concentration of bilirubin in artificial human serum as well. Figure 9. shows the CV peak current study of bilirubin in artificial serum utilizing the paper-based sensor.

## 4. Conclusions

Bilirubin is a residue of the body’s removal of senescent red blood cells that contain heme, which results in the breakdown of haemoglobin (Hb). High amounts of bilirubin in the bile and urine may indicate certain disorders. It is the cause of the jaundice’s yellow colouring. An essential test for assessing liver function is the evaluation of serum bilirubin in serum. Here, we developed a detection platform for Bilirubin by rationally implementing the extremely high charge-transfer efficiency of silver nanoparticles. Because paper is an inexpensive substrate and can be produced in large quantities, the use of electrochemical paper analytical device (EPAD) further enhances the sensor. A cost-effective platform for point-of-care diagnostics is provided by paper-based testing. These types of sensors are referred to as eco-designed analytical tools due to their efficient production, usage of the eco-friendly substrate, and potential to reduce waste management after measuring by incinerating the sensor. In this work, enzymes are used since they thought to be a unique and sensitive tool for use in rapid diagnostic methods. Cyclic voltammetry (CV) and linear sweep voltammetry (LSV), which were both validated with a potentiostat, were used to measure the analytical response of the biosensor. The proposed sensor shows a limit of detection about 1 μg/mL and linear range of about 1–9 μg/mL with good sensitivity. This eco-friendly and cost-effective approach could help in better and early detection of Bilirubin.

## Figures and Tables

**Figure 1 micromachines-13-01845-f001:**
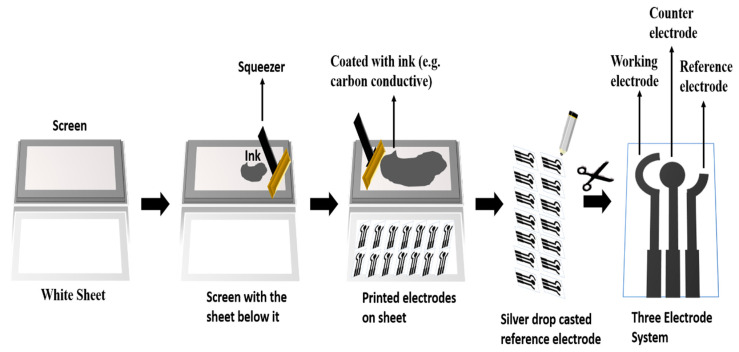
Schematic representation of construction of three electrode system-based Biosensor.

**Figure 2 micromachines-13-01845-f002:**
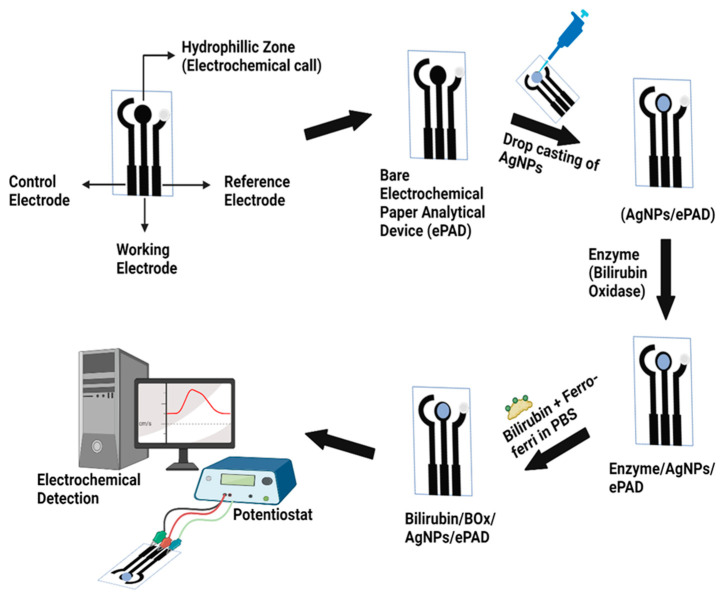
Schematic representation of working setup of Biosensor.

**Figure 3 micromachines-13-01845-f003:**
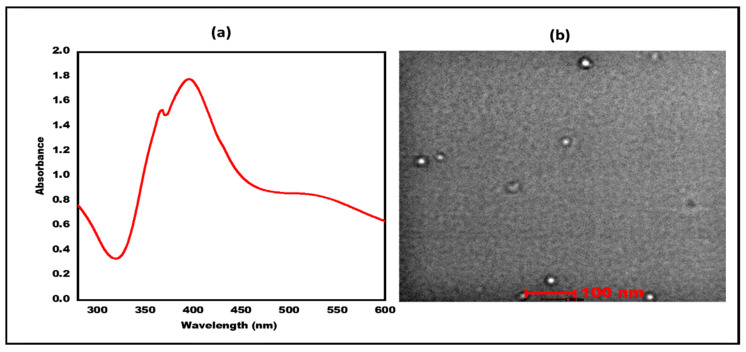
Characterization of Ag NPs (**a**) UV-Vis Spectroscopy (**b**) Transmission electron microscopy (TEM).

**Figure 4 micromachines-13-01845-f004:**
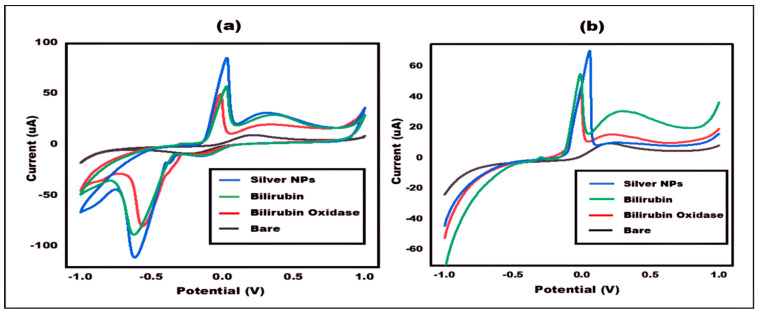
(**a**) Cyclic voltammetry of 10 mM PBS at bare ePAD, Ag-NPs/ePAD, Enzyme/Ag-NPs/ePAD, Bilirubin/enzyme/Ag-NPs/ePAD at 50 mVs^−1^ in the potential range from −1 V to +1 V. (**b**) LSV, in the range of −1 V to +1 V for 10 mM ferro/ferri in PBS at 50 mVs^−1^ at bare ePAD, Ag-NPs/ePAD, Enzyme/Ag-NPs/ePAD, Bilirubin/Enzyme/Ag-NPs/ePAD.

**Figure 5 micromachines-13-01845-f005:**
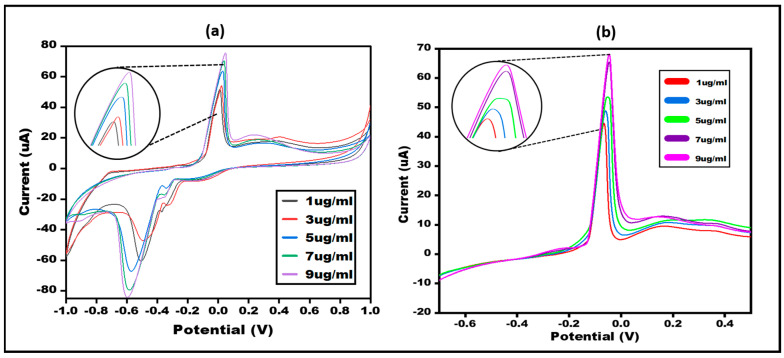
(**a**) CV of Bilirubin/Enzyme/Ag-NPs/PBs modified electrode using different concentrations of Bilirubin (1, 3, 5, 7, 9 µg/mL) in 10 mM PBS at 50 mVs^−1^ in the potential range from −1 V to +1 V. LSV of Bilirubin/enzyme/Ag-NPs/PBs modified electrode using different concentrations of Bilirubin (1, 3, 5, 7, 9 µg/mL) in10 mM PBS at 50 mVs^−1^ in the potential range from −1 V to +1 V. (**b**) LSV of Bilirubin/enzyme/Ag-NPs/PBs modified electrode using different concentrations of Bilirubin (1, 3, 5, 7, 9 µg/mL) in 10 mM PBS at 50 mVs^−1^ in the potential range from −1 V to +1 V.

**Figure 6 micromachines-13-01845-f006:**
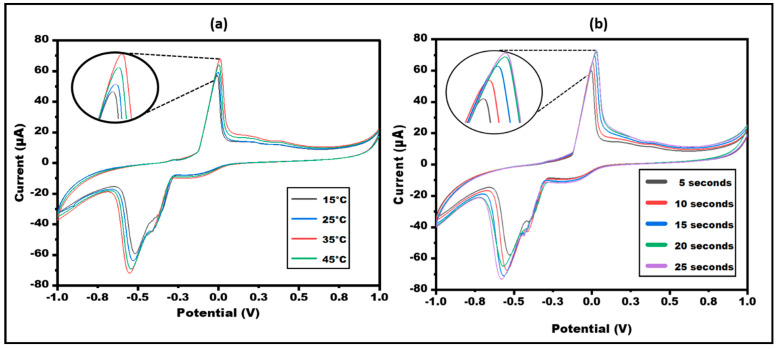
Cyclic voltammetry obtained at Bilirubin/Bilirubin oxidase/Ag NPs paper-based sensor for different (**a**) Temperature (15–45 °C) (**b**) Time (5–25 s) in 10 mM Ferro-Ferri in PBS at 50 mVs^−1^.

**Figure 7 micromachines-13-01845-f007:**
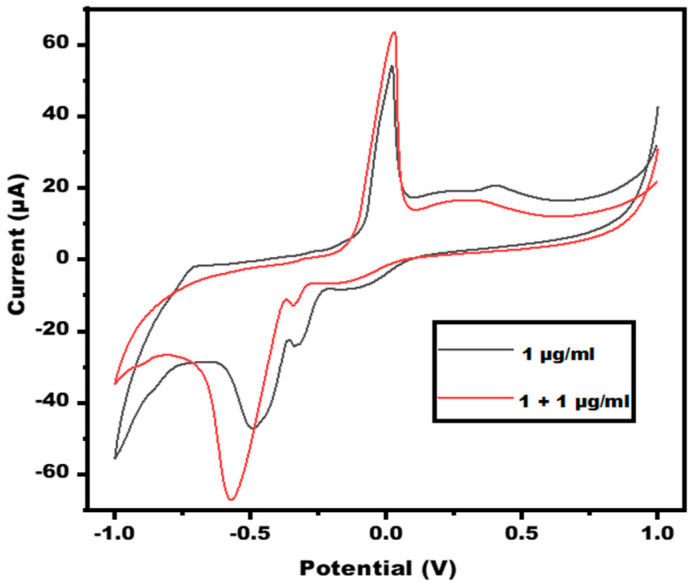
CV graph depicting the recovery value of Bilirubin/Enzyme/Ag-NPs Paper based sensor in 10 mM ferro-ferri in 0.1 M PBS at 50 mVs^−1^ in the potential range from −1 V to +1 V.

**Figure 8 micromachines-13-01845-f008:**
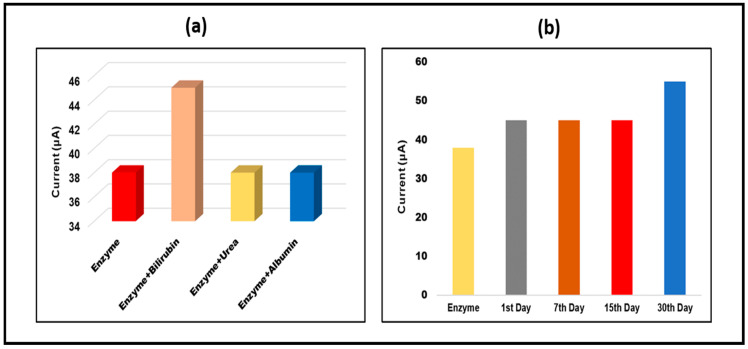
(**a**) Analysis of peak current CV-value of Bilirubin/Enzyme/Ag-NPs/PBs binding with the interferent, Urea and Albumin. (**b**) The electrochemical test was utilized to assess the stability of the constructed paper-based sensor for Bilirubin on the 1st, 7th, 15th, and 30th day.

**Figure 9 micromachines-13-01845-f009:**
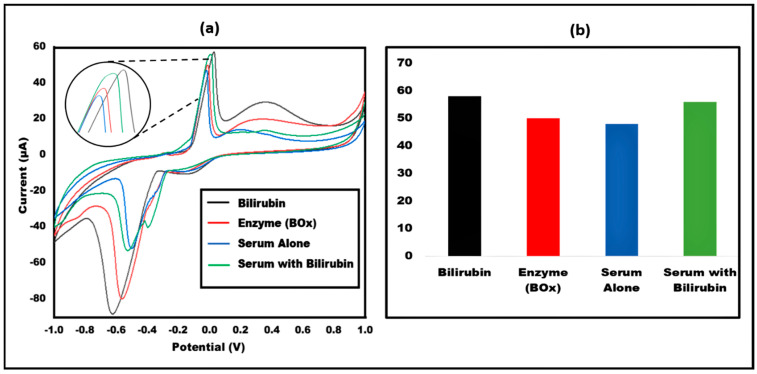
(**a**) Cyclic Voltammetry peak showing current study of Bilirubin in Artificial human serum utilizing fabricated paper-based sensor (**b**) Bar graph depicting the current study of Bilirubin compared to enzyme, Serum alone and Serum with bilirubin.

## Data Availability

Data can be available on request.

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
