# Peer review of "Electrochemical Sensor for Bilirubin Detection Using Paper-Based Screen-Printed Electrodes Functionalized with Silver Nanoparticles"

_micromachines, 2022, doi:10.3390/mi13111845_

Round 1
Reviewer 1 Report
Anzar et al. presented a study on a biosensor to check the serum biliribuine level, by exploiting the properties Ag nanoparticles. The properly presented the state of art and the aim of their work, then illustrated the protocol used to fabricate the biosensor and its potentiality.
I believe that the topic is really interesting and their contribute is consistent. The paper should be published after some few corrections related to the figures of the manuscript: Figures 1-3 and the graphical abstract are low resoluted. The TEM image should be more explicative and should have the scale bar.
Author Response
attached file

Reviewer 2 Report
Dear Authors,
The presented paper discusses a fundamental problem and fits the scope of the journal. A few concerns have to be addressed carefully:
- The reference number is not recommended since this is not the standard citation format. Please change it to 1,2,3, .......
- Graphical abstract is not recommended by MDPI style.
The experimental procedure was not clear. Please add more info describing the test setup in detail.
- Chemical reaction equation that appears on- page # 3 should have a similar font to that used in the text.
- Figures need re-demonstration such as (subscriptions).
- Merge relevant sections into a lumped and details section.
- The detection mechanism using the proposed sensor was not clear.
Author Response
attached file

Round 2
Reviewer 2 Report
Accepted.